# Disposition Kinetics of Amitraz in Lactating Does

**DOI:** 10.3390/molecules26164769

**Published:** 2021-08-06

**Authors:** Sathish Nanjundappa, Suresh Narayanan Nair, Darsana Udayan, Sreelekha Kanapadinchareveetil, Mathew Jacob, Reghu Ravindran, Sanis Juliet

**Affiliations:** 1Department of Veterinary Pharmacology and Toxicology, College of Veterinary and Animal Sciences, Kerala Veterinary and Animal Sciences University, Pookode, Lakkidi, P.O., Wayanad 673576, Kerala, India; drsathish642@gmail.com (S.N.); suresh@kvasu.ac.in (S.N.N.); darsanauday@rediffmail.com (D.U.); sreelekha.kp@gmail.com (S.K.); mathewnibu88@gmail.com (M.J.); sanis@kvasu.ac.in (S.J.); 2Department of Veterinary Parasitology, College of Veterinary and Animal Sciences, Kerala Veterinary and Animal Sciences University, Pookode, Lakkidi, P.O., Wayanad 673576, Kerala, India

**Keywords:** formamidine ectoparasiticide, dermal, blood concentration, milk, HPLC

## Abstract

Amitraz, a member of the formamidine pesticide family, commonly used for ectoparasite control, is applied as a dip or low-pressure hand spray to cattle and swine, and the neck collar on dogs. Data on amitraz were generated mainly on laboratory animals, hens, dogs, and baboons. The data on the toxicity and disposition of amitraz in animals and its residues in the milk are inadequate. Therefore, the present study was intended to analyze the disposition kinetics of amitraz and its pattern of elimination in the milk of lactating does after a single dermal application at a concentration of 0.25%. Blood at predetermined time intervals and milk twice daily were collected for eight days post application. The drug concentration was assayed by high-performance liquid chromatography (HPLC). Amitraz was detected in whole blood as early as 0.5 h, which attained a peak concentration at 12 ± 5 h, followed by a steady decline; however, detection persisted until 168 h. Amitraz was present in the blood at its 50% C_max_ even after 48 h, and was still detectable after 7 days. The disposition after a single dermal application was best described non-compartmentally. The mean terminal half-life (t_1/2_), mean residence time (MRT), and area under the curve (AUC_0–t_) were 111 ± 31 h, 168 ± 39 h, and 539 ± 211 µg/mL/h, respectively. The apparent volume of distribution (Vd_area_) was 92 ± 36 mL/g with an observed clearance (Cl) of 0.57 ± 0.33 mL/kg/h. Thus, the drug was well absorbed, widely distributed and slowly eliminated from the animal body. Amitraz achieved milk concentration approximating 0.2 per cent of the total dose after a single exposure and the steady-state elimination of amitraz in milk above the recommended maximum residue limit (MRL) of 0.01 mg/kg can act as a source of public health concern when applied on lactating animals.

## 1. Introduction

Amitraz, [1,5-di-(2,4-dimethyl phenyl)-3-methyl-1,3,5-triazapenta-1,4-diene] is a member of the formamidine pesticide family and has been used successfully to control ticks, mites, lice and many other pests on dogs, cattle, pigs, rabbits, and sheep [1,2]. Amitraz is the only drug approved by the Food and Drug Administration (FDA) for the treatment of generalized canine demodicosis [3]. Moreover, amitraz elicited high efficacy against one-host and multi-host ticks by inhibiting fecundity, preventing hatching of laid eggs, and by directly impairing the oocytes during vitellogenesis [4,5,6]. Its effectiveness is due to its alpha-adrenergic agonist activity, interaction with octopamine receptors of the central nervous system, and inhibition of monoamine oxidases and prostaglandin synthesis [7,8,9,10].

Goats are among the main meat-producing animals in India, whose meat (chevon) is one of the choicest meats and has a huge domestic demand. Besides meat, goats provide other products such as milk, skin, fiber, and manure [11]. Among diseases affecting ruminants, ticks and tick-borne diseases are ranked high in terms of their impact on the livelihood of poor farming communities in developing countries [12,13]. The incidence and prevalence of tick infestation in goats are reported from different regions of the country [14,15]. Currently, amitraz is one of the chemical acaricides recommended for goats, used topically in the form of spray, pour on, or bath. The toxicity and metabolic fate of amitraz in rats, hens, baboons, lactating dairy cows, pigs, and humans are well documented [9,10,16,17]. Likewise, reports on the toxicokinetics of amitraz in dogs, following oral administration, and sheep and ponies, following intravenous administration, are also available [18,19]. Recently, a study revealed that the central nervous system toxicity subsequent to systemic and brain exposure of amitraz and its major metabolites in rats were mainly attributed to its related metabolite BTS27271 exposure in the brain tissues [20]. Still, adequate information with respect to the toxicokinetic behavior of amitraz in goats following dermal application is lacking. Moreover, many incidences of amitraz poisoning in humans were reported in journals [21]. 

The present study therefore analyzes the disposition kinetics of amitraz in lactating does, and its pattern of elimination in milk following a single dermal application at the rate of 0.25%. 

## 2. Results

Following a single dermal application of amitraz at a dose of ten times the recommended concentration, the animals did not reveal any signs of toxicity during the entire duration of the study period. All goats showed normal behavior and appetite.

Amitraz was detected at a wavelength of 260 nm. The calibration curve for amitraz, prepared using linear regression analysis, fitted well over the range of 10–100 µg/mL with a regression coefficient (R^2^) of 0.986. The regression formula obtained from the calibration curve was further used to quantify the concentration of amitraz in analyzed samples. The limit of detection (LOD) and limit of quantification (LOQ) for amitraz were 0.40 µg/mL and 1.35 µg/mL, respectively. The detection limit was evaluated by the peak signal/noise (S/N) ratio. An S/N greater than 3 was considered a detectable peak. Maximum recovery of amitraz was observed after fortifying blood and milk with 10, 50, and 100 µg/mL of amitraz with methanol as the extraction solvent (>85% recovery for blood and >95% recovery for milk). The chromatogram for amitraz recovered from the blood and milk, following fortification with a known quantity, is depicted in Figure 1 and Figure 2. 

The result of plasma concentration after dermal application of amitraz is given in Figure 3. It was noticed that after dermal application, amitraz entered into the bloodstream within half an hour and achieved a concentration of 2.73 µg/mL. From then onwards, the plasma concentration steadily elevated to 7.68 µg/mL (C_max_) and then started declining at a slow pace. Interestingly, the drug was present at its 50% C_max_ even after 48 h and was still detectable after 7 days.

Toxicokinetic analysis of amitraz was performed using PKSolver and the results of the analysis are given in Table 1. Since the mathematical fidelity of the compartmental models was low, a non-compartmental analysis was performed on the plasma concentration data. It was observed that after dermal application, a peak plasma concentration was observed at 12 ± 5 h, which was then followed by an initial steep decline and then a slower elimination from the plasma. The mean t_1/2_ of amitraz was as high as 111 ± 31 h, and the mean residence time was 168 ± 39 h. The mean AUC_0–t_ was 539 ± 211 µg/mL·h while the AUC_0–∞_ was 874 ± 254 µg/mL·h. The drug appears to be well distributed in the tissue, as indicated by a high plasma apparent volume of distribution of 92 ± 36 mL/g, and is eliminated by an observed clearance of 0.57 ± 0.33 mL/kg/h.

The mean concentration of amitraz in the milk, collected twice daily in the morning and evening up to eight days after single dermal administration, are presented in Figure 4. Amitraz could be detected in milk from 12 h post-dermal application and for the remaining duration of the study period of eight days. The concentration of amitraz did not significantly vary between the milk collected in the morning and evening. The concentration ranged from 0.37± 0.03 µg/mL to 0.44 ± 0.03 µg/mL. Additionally, it was also observed that the concentration did not decline even at the end of the experiment, which is a major public health concern. The total concentration excreted in the milk accounted for almost 0.2 percent of the total dose.

## 3. Discussion

Reports on incidences of amitraz poisoning in animals and humans have increased in recent years due to its increased production and use [22,23,24]. Amitraz poisoning can occur by inhalation, ingestion, and skin contact. Due to its α_2_-agonistic action, the most common adverse effects associated with systemic amitraz toxicity are central nervous system alterations, such as depression, hypotension, polyuria, hyperglycemia, vomiting, and respiratory failure [9,10,17,25,26,27]. 

Even though the applied concentration of the drug was ten times more than the normal therapeutic concentration, no toxic signs were observed in the present study. All the goats exhibited normal behavior and appetite. The results were consistent with those from several previous studies evaluating the dermal toxicity of amitraz. Neither a single dermal application of 1600 mg/kg in rats and 0.5 g in New Zealand white rabbits, nor repeated dermal exposure (two-week intervals) of 500 mg amitraz in guinea pigs, revealed any local or systemic effects [16]. However, more marked and persistent dermal reactions were reported in guinea pigs following intradermal injections of 5% (*w*/*w*) amitraz in Alembicol D (coconut oil) twice within a two-week interval [16]. 

Amitraz was rapidly absorbed into the systemic circulation within 30 min, achieving a peak concentration at 12 ± 5 h that persisted for 168 h post-dermal application. Nevertheless, earlier reports stating a peak blood concentration within 24–72 h of application of 20–21 mg ^14^C-amitraz on beagle dogs are available [16]. On the other hand, present findings contradict the previous report of DeLay et al. [28], that amitraz could not be detected in the plasma over 56 h following topical application of a spot-on formulation containing a combination of amitraz and metaflumizone at the rate of 20 mg/kg each on the dogs. This could be due to the difference in the dosage forms used in the above studies and their rate of dissolution. Additionally, studies conducted by Reveire et al. [29] demonstrated that the formulation components, and not exclusively the physical–chemical properties of the penetrant, were the major factors in the assessment of absorption and permeation of topical ectoparasiticides.

Furthermore, previous studies conducted in rats, beagles, and pigs following dermal treatment of ^14^C-amitraz showed that the washing of treated skin with soap and water 10–12 h after application resulted in the removal of 60–90% of the applied amitraz, with only 3% and 1.4% of the drug remaining on the skin after 24 h and after 5 days, respectively. Hence, it is presumed that a portion of the applied drug that remained on the skin and absorbed into the systemic circulation might have attributed to its detection in the blood until 168 h. Following dermal administration, 90% of the drug was eliminated in the urine and feces, with very low concentration (less than 0.05 ppm) found in most of the tissues [16,30]. Since recovery and tissue residual concentrations were not carried out in the present study, the proportion excreted through urine, feces, and the amount retained in the tissues could not be quantified. 

Detection of amitraz in the blood half an hour after the dermal application itself showed that the compound was quickly absorbed through the cutaneous route. At 0.5 h, it had achieved 35.5% of the maximum plasma concentration (C_max_), signifying its rapid rate of cutaneous absorption. This is also reinforced by a high value of absorption half-life (by two-compartmental analyses, data not shown) of 0.95 ± 0.19 h. The plasma concentration declined quickly until 24 h, which may be due to the distribution of the drug rather than its elimination, which were concluded from the following observations. The apparent volume of distribution of the drug was high, indicating the movement of the drug from the central to the peripheral compartment rather than being retained in the plasma. A similar observation was obtained with one compartmental analysis (data not shown) where the rate constant of peripheral to central compartment was much less (0.01 ± 0.00/h). The above two findings, along with a lower value of observed clearance and persistent presence of amitraz in milk at constant levels for 8 days, show that the drug undergoes extensive distribution within the body. The ratio of the AUC_0–t_ to the AUC_0–α_ suggests that even by day 7, only 61.7% of the drug was eliminated. This, along with a high MRT of 168 ± 39 h, proves the proclivity of amitraz for high tissue retention and probable tissue residues of amitraz following dermal application. This protracted period of retention may be attributable to a high applied dose (10 times the clinically recommended dose), although the same toxicokinetic pattern is expected to be elicited with clinical doses too. The major contributory factor for prolonged half-life can be a delayed clearance, high volume of distribution and/or extravascular migration of the drug, which all are dependent on the lipophilicity of the drug. Amitraz shares a common property of being highly lipophilic with other classes of acaricides having an octanol/water partitioning coefficient (Log P) value of 5.5 [31,32]. This is further supported by the persistent level of amitraz detected in the milk. It appears in the milk quickly after the dermal absorption.

Milk, being a medium containing fat, forms an excellent route of elimination of lipophilic drugs, which could be the reason for its quick appearance in the milk. In addition, the longer persistence of the drug in the milk at almost constant levels revealed that the drug is retained in tissue stores, and slowly re-enters the systemic circulation, since it is lipophilic, for its elimination through the milk. These findings were consistent with the studies conducted in a cow following oral administration of amitraz in capsules [16]. Moreover, amitraz was detected at a concentration of 0.01 mg/kg in the milk of cattle dipped with Taktic^®^ (0.025 per cent of amitraz) [30].

In addition, dipping of heifers in different concentrations of amitraz (Taktic^®^ 12.5 per cent EC) protected against re-infestation by *Rhipicephalus microplus* larvae for 14 days [33]. A formulation containing amitraz and metaflumizone also protected against re-infestation by fleas (*Ctenocephalides felis*) and ticks (*R. sanguineus, Dermacentor variabilis, Ixodes scapularis, Amblyomma americanum*) for six and four weeks, respectively, on dogs [34]. Similarly, spot-on formulation containing fipronil, amitraz, and (S)-methoprene protected against *C. canis* on dogs for seven weeks [35]. The efficacy of amitraz observed in the above reports could be correlated to the prolonged persistence and slow elimination of the drug from the body.

## 4. Materials and Methods

All the solvents and chemicals used in this study were procured from M/s Merck India Ltd., Mumbai, India and M/s Sigma-Aldrich India Ltd., Bangalore, India. Amitraz (Dr. Ehrenstorfer GmbH, Augsburg, Germany, C 10230000 with purity ≥ 99%) was procured from M/s Merck, Darmstadt, Germany and used as the HPLC standard. Water used in all experiments was purified on a Milli-Q^®^ system from Millipore (Bedford, MA, USA).

To determine the amitraz concentration, a stock solution of amitraz at a concentration of 100 µg/mL in acetonitrile was prepared and serially diluted to final concentrations of 10, 20, 30, 40, 50, 60, 70, 80, and 90 µg/mL. These were stored at −20 °C in the dark.

Six healthy adult female lactating Malabari goats of ages 1.5–2 years, weighing between 15–20 kg body weight, were procured from the Instructional Livestock Farm, College of Veterinary and Animal Sciences, Pookode and were used in the present study. All goats were kept in individual cages and fed a standard concentrate ration (satisfying 14–16% DCP and 17% TDN), green fodder (3 kg of Congo signal grass) and water *ad libitum*. Before the start of the experiment, animals were acclimatized in the cages and then kept under observation for a clearance time of seven days. During this period, the body temperature, respiration rate, heart rate, rumen motility, defecation, and urination were recorded daily. The entire research protocol was approved by the Institutional Animal Ethics Committee of College of Veterinary and Animal Sciences, Pookode. 

The high-performance liquid chromatography system (HPLC) (Shimadzu Corporation, Tokyo, Japan) equipped with LC-10AT quaternary gradient pump, Rheodyne manual loop injector 20 µL, column oven CTO-10AS VP, diode array detector SPD-M20A and LC-solutions version software for data analysis were used. The separation was performed on Phenomenex Luna 5U C18 column (250 × 4.6 mm i.d.) 

Amitraz was detected using the mobile phase acetonitrile:methanol:formic acid (0.1%) at 30:10:60, respectively. The flow rate was maintained at 0.5 mL/min. The oven temperature was 50 °C and the injection volume was 20 µL. The wavelength of the detector was set at 250–300 nm to detect amitraz. Ascending standards of amitraz at 10, 20, 30, 40, 50, 60, 70, 80, 90, and 100 µg/mL in acetonitrile were analyzed using HPLC and peaks with their area were calculated. The reproducibility of the results was verified at least thrice with each concentration of the amitraz. The calibration curve of amitraz concentration versus time was plotted on graph paper. Linear regression was performed on the data set. The limit of detection (LOD) and limit of quantification (LOQ) were also determined. 

Blood and milk were (10 mL each) collected and recovery of the amitraz from the blood and milk was carried out to ascertain the reliability of the method after fortifying the blood and milk with amitraz standards at 10, 50, and 100 ppm concentrations. With blood and milk being highly complex matrices, the precision and accuracy of the extraction methods were determined in order to ensure minimum matrix interference. Three replicates were used for each concentration. After the necessary workup, the concentration of amitraz from the blood and milk samples were analyzed by HPLC. The extraction efficiency was determined for amitraz by comparing the peak areas from drug free samples spiked with a known quantity of drug in the range of concentration of the calibration curves and standard solutions in the mobile phase, injected directly into the analytical column. The mean and standard deviation of each concentration were calculated. 

For the dermal application, amitraz 0.25% (10 times more than the recommended therapeutic concentration) was prepared from the commercially available product (RIDD^®^) by dissolving 2 mL solution into 98 mL of water. A strip of skin (300 mm long and 150 mm wide) covering the vertebral spine in the mid-line of each goat was shaved and cleaned from the withers to the lumbar area 24 h before the application of amitraz. The prepared concentration of the drug was applied to the shaved area of the animals using a manually operated sprayer. 

Before the start of the experiment, the jugular fossa of the neck of each goat was shaved and cleaned with a tincture of iodine. Blood samples (2.5 mL) were collected from the right jugular venipuncture into the heparinized tubes just before the treatment and subsequently at 0.5, 1, 2, 4, 6, 12, 24, 36, 48, 72, 96, 120, 144, and 168 h post-application of the drug. This was according to the method in [36] with slight modifications. Milk samples from the lactating does were collected by hand milking into 10 mL clean and sterilized vials before treatment and during milking time in the morning and evening every day post-application for up to eight days. The samples were stored at −20 °C until they were processed for analysis by HPLC.

The collected blood samples (1 mL) were deproteinized by adding 2.5 mL of saturated ammonium sulfate in 2.5% sulphuric acid (H_2_SO_4_) and 2 mL methanol. The sample was mixed thoroughly and the supernatant was filtered through sodium sulphate (~0.5 g). The extraction was repeated with methanol, the supernatant was pooled, and the volume was reduced using a rotary vacuum evaporator (M/s Buchi, Flawil, Switzerland) set at 40 °C and 337 mbar. The final volume was made up to 1 mL with acetonitrile (Lichrosolv^®^). The resulting sample was filtered through a 0.2 µm Millipore filtering unit for subsequent HPLC analysis. This was performed as per the protocol described in the literature [37] with minor modifications. The processed sample (20 µL) was injected into the HPLC for the quantitation of the amitraz.

Similarly, 1 mL of milk was placed in centrifuge tubes. To precipitate the milk proteins, 2.5 mL of saturated ammonium sulfate in 2.5% sulphuric acid (H_2_SO_4_) and 2 mL of methanol were added to it. The sample was mixed thoroughly and centrifuged at 6000 rpm for 15 min. The supernatant was filtered through sodium sulphate (~0.5 g). The procedure was repeated three times. The supernatant thus collected was pooled and the volume was reduced using a rotary vacuum evaporator (M/s Buchi, Switzerland) set at 40 °C and 337 mbar. The final volume was made up to 1 mL with acetonitrile and filtered through a 0.2 µm Millipore filtering unit for subsequent HPLC analysis. The processed sample (20 µL) was injected into the HPLC for the quantitation of the amitraz.

The blood levels of amitraz were plotted on a semi-logarithmic paper against time. The concentrations of amitraz in each animal at 5, 10, 15, 30, 45 min and 1, 1.5, 2, 4, 6, 8, 12, 24, 36, 48, 72, 96, 120, 144, and 168 h were analyzed non-compartmentally [38] using PKsolver software [39]. The mean and standard error of each kinetic parameter were determined [40]. The concentration of amitraz in the milk was directly calculated from the HPLC chromatogram and was represented graphically.

## 5. Conclusions

The present study was conducted to find out the disposition kinetics of the commonly used insecticide amitraz in lactating Malabari does after dermal application, and the determination of the levels of its residue in the does’ milk. It was demonstrated that goats with topically applied amitraz at ten times the recommended therapeutic dose did not show any adverse toxic effects. Amitraz was quickly absorbed into the systemic circulation after cutaneous application and it persisted in the blood for a prolonged period of time. Furthermore, the low value of clearance, high volume of distribution, and high MRT substantiated the tendency of amitraz for high tissue retention and probable tissue residues following dermal application. Its persistence and elimination in the milk for an extended period are of public health significance. 

## Figures and Tables

**Figure 1 molecules-26-04769-f001:**
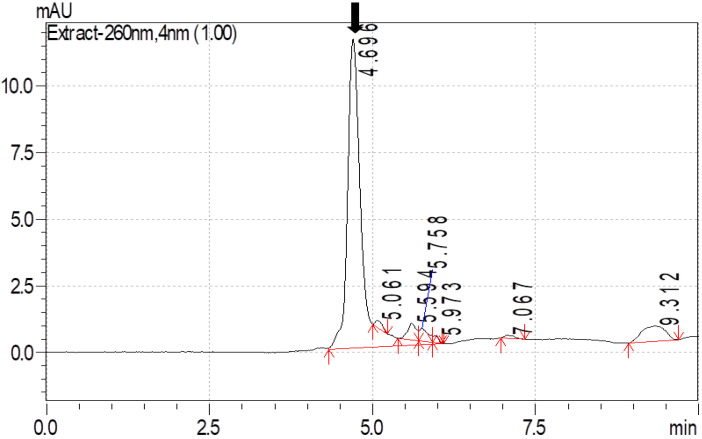
Chromatogram of amitraz recovered from the blood following fortification with a known quantity.

**Figure 2 molecules-26-04769-f002:**
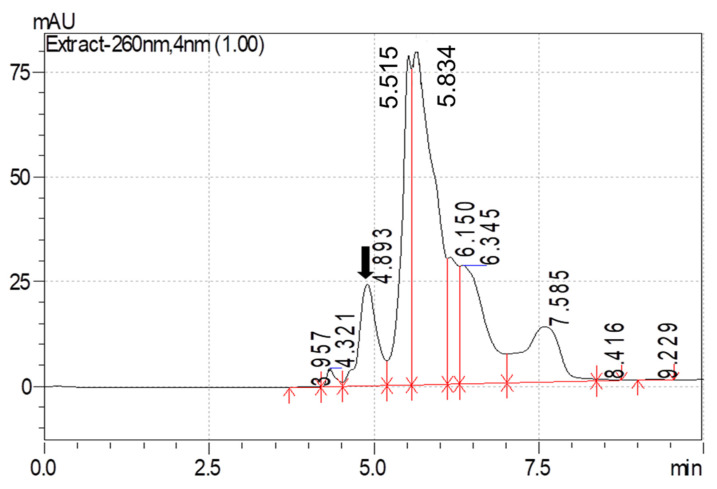
Chromatogram of amitraz recovered from the milk following fortification with a known quantity.

**Figure 3 molecules-26-04769-f003:**
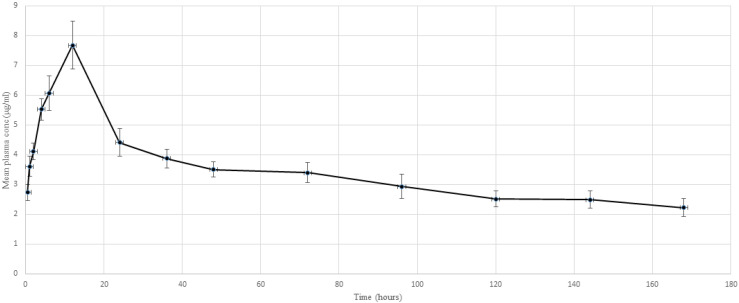
Semi logarithmic plot of the mean blood concentration (μg mL^−1^) of amitraz against time following a single dermal application at 0.25 percent in lactating does. (*n* = 6).

**Figure 4 molecules-26-04769-f004:**
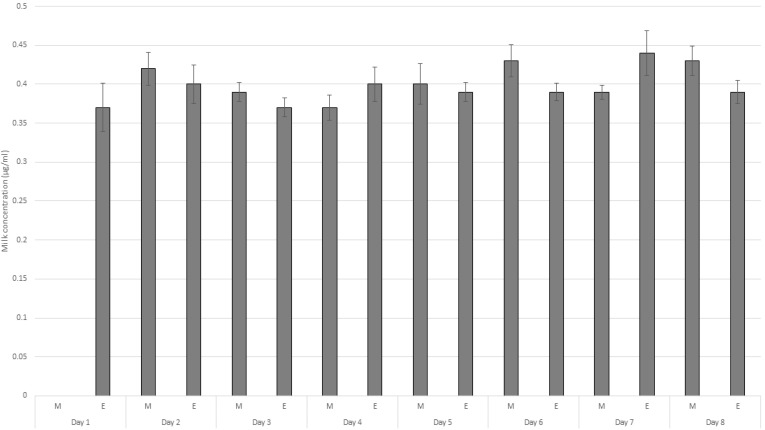
Plot of the mean concentration of amitraz in milk collected twice daily in the morning and evening up to eight days after single dermal administration (*n* = 6, mean ± SE values; M = morning, E = evening).

**Table 1 molecules-26-04769-t001:** Toxicokinetic parameters for amitraz calculated from the mean blood concentration after a single dermal application to Malabari goats.

Parameters	Units	Mean ± SE
λ_z_	1/h	0.006 ± 0.004
t_1/2 λz_	h	111 ± 31
T_max_	h	12 ± 5
C_max_	μg/mL	8 ± 3
C_last_/C_max_		0.26 ± 0.05
AUC_0–t_	μg/mL·h	539 ± 211
AUC_0–∞_	μg/mL·h	874 ± 254
AUC_0–t/0–∞_		0.62 ± 0.34
AUMC_0__–__inf_	μg/mL·h^2^	146,808 ± 24,363
MRT_0__–__inf_	h	168 ± 39
Vd_ss_	mL/gm	92 ± 36
Cl	mL/kg/h	0.57 ± 0.33

Pharmacokinetic parameters are calculated with PKSolver MS_Excel add in. (*n* = 6); λ_z_—the terminal slope of a semilogarithmic concentration–time curve; t_1/2 λz_—terminal half-life; T_max_—time of maximum blood concentration; C_max_—maximum blood concentration; C_last_/C_max_—the ratio of the last blood concentration and maximum blood concentration; AUC_0–t_—the area under the curve from zero to time t; AUC_0__–__inf_—the area under the curve from zero to infinity; AUMC_0–inf_—the area under the first-moment curve; MRT_0–inf_—the mean residence time; V_ss_—the apparent volume of distribution at steady-state; Cl—the total body clearance of the drug from the blood.

## Data Availability

The data presented in this study are available within the article.

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
