# Peer review of "Disposition Kinetics of Amitraz in Lactating Does"

_molecules, 2021, doi:10.3390/molecules26164769_

Round 1
Reviewer 1 Report
The manuscript investigates the elimination kinetics of amitraz in blood and milk of goats after a single dermal application. The topic is of interest given that amitraz is a frequently used pharmaceutical for animals and public health concerns can be raised if the compound persists in milk and subsequently in dairy products.
I think that the manuscript is interesting and can be accepted for publication in the journal after minor revision:
- The manuscript has a decent analytical explanation of the sample preparation procedure. However, I would like to see some more text describing the quality assurance and quality control that the authors used to assure their measurements.
- One very important comment is that the authors must be careful with significant digits. For example, 110.88 ±31.114 h. Do you have so high accuracy? I would write this number as 111±31 h. The same comment applies for all numbers in the manuscript (indicative examples: 0.44 ± 0.029, 0.37± 0.031μg/mL
- Figure 1 and Figure 2 do not carry much information. I would eliminate them. Moreover, Figure 2 shows a relatively bad linearity. What can be the cause of this fact?
- The section “Conclusion” can be enlarged a little bit. It looks very poor right now
Author Response
- The manuscript has a decent analytical explanation of the sample preparation procedure. However, I would like to see some more text describing the quality assurance and quality control that the authors used to assure their measurements.
Response: Thank you for the suggestion. The text describing the quality assurance and quality control that we used are incorporated in the material methods in the revised manuscript.
“The reproducibility of the results was verified at least thrice with each concentration of the amitraz.”
“Blood and milk being highly complex matrices, the precision and accuracy of the extraction methods were determined in order to ensure minimum matrix interference.”
“The extraction efficiency was determined for amitraz by comparing the peak areas from drug free samples spiked with known quantity of drug in the range of concentration of calibration curves and standard solutions in the mobile phase, injected directly into the analytical column.”
- One very important comment is that the authors must be careful with significant digits. For example, 110.88 ±31.114 h. Do you have so high accuracy? I would write this number as 111±31 h. The same comment applies for all numbers in the manuscript (indicative examples: 0.44 ± 0.029, 0.37± 0.031μg/mL
Response: In the present study, we used six healthy adult female lactating Malabari goats. The results shown were Mean and standard error values of the pharmacokinetic parameters calculated from six individual goats. As suggested by the learned reviewer, the corrections accordingly are done in the revised manuscript.
- Figure 1 and Figure 2 do not carry much information. I would eliminate them.
Response: We have removed figure 1 and figure 2 from the manuscript.
Moreover, Figure 2 shows a relatively bad linearity. What can be the cause of this fact?
Response: We had plotted the data as obtained after calculating the mean. In addition, we have included the chromatogram of blood and milk sample in the revised manuscript.
- The section “Conclusion” can be enlarged a little bit. It looks very poor right now
Response: The conclusion is modified as suggested
Present study was conducted to find out the disposition kinetics of the commonly used insecticide, amitraz in lactating Malabari does, after dermal application and determination of levels of its residue in milk. It was demonstrated that goats applied topically with amitraz at ten times the recommended therapeutic dose did not show any adverse toxic effects. Amitraz
was quickly absorbed into the systemic circulation after cutaneous application and it persisted in the blood for a prolonged period of time. Further, the low value of clearance, high volume of distribution, and high MRT substantiated the tendency of amitraz for high tissue retention and probable tissue residues following dermal application. Its persistence and elimination in milk for an extended period are of public health significance.
Reviewer 2 Report
Although the presented material does not have a novel approach, it is a correctly conducted pharmacokinetic study whose results may be interesting in terms of the species studied and the toxicological risk presented by acaricide amitraz. The study results and their interpretation are briefly and clearly presents.
As an observation, the chromatogram of a standard solution and the graph of the calibration curve should be included in additional materials. At most one chromatogram of a real blood or milk sample could be included in the manuscript.
In the section ‘’materials and methods’’ describing HPLC analysis, the issue of matrix interferences should be discussed, and from this point of view the chromatogram of blood or milk samples would be interesting.
Another unclear point is how to calculate the amitraz concentration in the analyzed samples. Given that, the calibration curve was plotted for standard solutions in acetonitrile, was the recovery coefficient considered for the concentration calculation?
Author Response
- As an observation, the chromatogram of a standard solution and the graph of the calibration curve should be included in additional materials. At most one chromatogram of a real blood or milk sample could be included in the manuscript.
Response: As per the suggestion of the first reviewer, the figures 1 &2 were removed. We included the chromatogram of blood and milk sample in the revised manuscript as figures1 and 2.
- In the section ‘’materials and methods’’ describing HPLC analysis, the issue of matrix interferences should be discussed, and from this point of view the chromatogram of blood or milk samples would be interesting.
Done as suggested. “Blood and milk being highly complex matrices, the precision and accuracy of the extraction methods were determined in order to ensure minimum matrix interference.”
- Another unclear point is how to calculate the amitraz concentration in the analysed samples.
In the present study, the regression formula obtained from the calibration curve was used to quantify the concentration of amitraz in analysed samples.
Y = mX + C; Where, Y = peak area, C = Y intercept, m = Slope of the calibration curve, X= Concentration (μg/mL).
Incorporated in both Results section and Materials and methods section.
Given that, the calibration curve was plotted for standard solutions in acetonitrile, was the recovery coefficient considered for the concentration calculation?
Response: Not considered for the calculation of concentration.
Round 2
Reviewer 2 Report
Seems that authors considered all the sugestion in the revision of the manuscrispt. As an observation:
the paragraph:
''The regression formula (Y = mX + C; Where, Y = peak area, C = Y intercept, m = Slope 240 of the calibration curve, X= Concentration in μg/mL) derived from the calibration curve 241 was used to quantify the concentration of amitraz in the analyzed samples.'' (R240-241)
is redundant. My previous observation refers to the calculation of the amitraz concentration with or without the recovery coefficient. In my opinion, it should be included in the calculation for reliable results.
Author Response
Response to reviewer comments
- Seems that authors considered all the suggestion in the revision of the manuscript. As an observation: in the paragraph: ''The regression formula (Y = mX + C; Where, Y = peak area, C = Y intercept, m = Slope 240 of the calibration curve, X= Concentration in μg/mL) derived from the calibration curve 241 was used to quantify the concentration of amitraz in the analyzed samples.'' (R240-241) is redundant.
Response: In the revised manuscript, we deleted the statement “''The regression formula (Y = mX + C; Where, Y = peak area, C = Y intercept, m = Slope of the calibration curve, X= Concentration in μg/mL) derived from the calibration curve was used to quantify the concentration of amitraz in the analyzed samples.''
- My previous observation refers to the calculation of the amitraz concentration with or without the recovery coefficient. In my opinion, it should be included in the calculation for reliable results.
Response: We calculated the amitraz concentration in the samples taking the recovery into consideration.